Continued nucleic acid tests for SARS-CoV-2 following discharge of patients with COVID-19 in Lu’an, China

Lyu Yong lyong@lacdc.com.cn 1
Wang Danni 2
Li Xiude 1 3
Gong Tianqi 1
Xu Pengpeng 1
Liu Lei 1
Sun Jie 1
1 Lu’an Municipal Center for Disease Control and Prevention , Lu’an , China
2 Teaching Center for Preventive Medicine, School of Public Health, Anhui Medical University , Hefei , China
3 Department of Nutrition and Food Hygiene, School of Public Health, Anhui Medical University , Anhui Hefei , China
Sotelo-Mundo Rogerio
Electronic publication date: 2021 Jun 22
Publication date: 2021
Volume: 9
Electronic Location ID: e11617
Received 2020 Aug 10; Accepted 2021 May 25
Copyright: ©2021 Lyu et al.
Copyright year: 2021
Copyright holder: Lyu et al.
License: This is an open access article distributed under the terms of the Creative Commons Attribution License, which permits unrestricted use, distribution, reproduction and adaptation in any medium and for any purpose provided that it is properly attributed. For attribution, the original author(s), title, publication source (PeerJ) and either DOI or URL of the article must be cited.
License URL: https://creativecommons.org/licenses/by/4.0/

Keywords: COVID-19, SARS-CoV-2, RT-PCR, Discharge criteria

Funding: The authors received no funding for this work.

==============================
Background

Studies have shown that discharged Coronavirus disease 2019 (COVID-19) patients have retested positive for SARS-CoV-2 during a follow-up RT-PCR test. We sought to assess the results of continued nucleic acid testing for SARS-CoV-2 patients in COVID-19 patients after they were discharged in Lu’an, China.

Methods

We conducted RT-PCR tests on sputum, throat swabs, fecal or anal swabs, and urine samples collected from 67 COVID-19 patients following discharge. Samples were collected on the 7th and 14th days following discharge. Patients testing positive on the 7th or 14th day were retested after 24 hours until they tested negative twice.

Results

Seventeen (17/67, 25.4%) discharged COVID-19 patients had a positive RT-PCR retest for SARS-CoV-2. Among them, 14 (82.4%) were sputum positive, five (29.4%) were throat swab positive, seven (41.2%) were fecal or anal swab positive, one (5.9%) was urine sample positive, five (29.4%) were both sputum and throat swab positive, four (23.5%) were both sputum and fecal test positive, and one (5.9%) was positive of all four specimens. The shortest period of time between discharge and the last positive test was 7 days, the longest was 48 days, and the median was 16 days. The proportion of positive fecal or anal swab tests increased from the third week. The median Cq cut-off values after onset were 26.7 after the first week, 37.7 the second to sixth week, and 40 after the sixth week. There were no significant differences between the RT-PCR retest positive group and the unrecovered positive group.

Conclusions

There was a high proportion of patients who retested positive for COVID-19. Discharge criteria have remained fairly consistent so we encourage regions affected by COVID-19 to appropriately amend their current criteria.

Introduction

Coronavirus disease 2019 (COVID-19), which is caused by the novel severe acute respiratory syndrome coronavirus 2 (SARS-CoV-2), has spread rapidly since December 2019 (Li et al., 2020b; Zhou et al., 2020b). There have been a total of 85,557 confirmed COVID-19 cases in mainland China as of October 10, 2020. 94.3% (80,705/85,558) of these cases recovered and were discharged from the hospital (National Health Commission of the People’s Republic of China, 2020a). The high recovery rate improved public confidence in the government’s response to the emergency; however, the situation around the world remains grim. As of October 15 2020, there have been 38,202,956 confirmed cases of COVID-19, including 1,087,069 deaths, reported to the World Health Organization (WHO), indicating that this pandemic cannot be ignored, especially in this era of globalization (World Health Organization, WHO, 2020a, https://covid19.who.int/).

COVID-19 patients discharged in China were RT-PCR tested and found to be positive for SARS-CoV-2 again during follow-up care (China News Weekly, 2020; Dou et al., 2020; Lan et al., 2020; Li et al., 2020a; Tang et al., 2020; Xiao, Tong & Zhang, 2020; Xing et al., 2020; Zhang et al., 2020; Zheng et al., 2020). Other countries have also reported patients that were positive when retested (Abdullah et al., 2020; Cento et al., 2020; Hartman, Hess & Connor, 2020; Zanardini et al., 2020). These studies have not shown that patients who retested positively were contagious (Wang, 2020), however, it has created public concern.

The World Health Organization (WHO) has updated its criteria for discharging isolated COVID-19 patients; in this criteria retesting is not required (World Health Organization, WHO, 2020b, https://www.who.int/news-room/commentaries/detail/criteria-for-releasing-covid-19-patients-from-isolation ; World Health Organization, WHO, 2021, https://www.who.int/publications/i/item/clinical-management-of-covid-19). Countries have also updated their discharge protocols. For example, according to the new Korea Centers for Disease Control (KCDC) discharge protocols, “no additional tests are required for cases that have been discharged from isolation” (Korea Centers for Disease Control, KCDC, 2020). However, there is merit to retesting discharged cases and additional study is required to re-evaluate the current discharge criteria. We present the results of continued nucleic acid testing following the discharge of patients with COVID-19 in Lu’an, China.

Materials & Methods

Setting and samples

There have been a total of 69 confirmed COVID-19 cases in Lu’an, China, as of April 16, 2020, with the first case confirmed on January 22, 2020. The first case was discharged from the hospital on January 30, 2020 and the last case was discharged on March 3, 2020. Two COVID-19 cases left Lu’an after being discharged from the hospital. Therefore, nucleic acid tests were conducted for 67 cases.

Discharge criteria

The Chinese discharge criteria for COVID-19 patients have been consistent between the second and the seventh iterations and are based on following criteria: 1) normal temperature (<37.3 °C) for 3 days; 2) reduced respiratory tract symptoms and clear clinical improvement confirmed by pulmonary imaging, and 3) collection of two consecutive negative RT-PCR tests on respiratory samples at least 24 h apart (National Health Commission of the People’s Republic of China, 2020b).

Sample collection methods

An epidemiological investigation was conducted and all 67 patients were confirmed to have maintained distance from others and remained indoors after discharge, thus ruling out possible reinfection. Sputum, throat, fecal or anal swabs, and urine samples were collected on the 7th and 14th day following discharge. If a patient tested negative, a second test was performed more than 24 h after the first test until the patient was confirmed to be negative for two consecutive tests. The patient was immediately required to continue the quarantine protocol in the hospital for at least 14 days.

Nucleic acid test strategy

RT-PCR was used to conduct nucleic acid testing. The RT-PCR kit was manufactured by Sun Yat-sen University Da’an Gene Co., Ltd., China (20203400063). The ROC curve was used to determine the ORF1 ab and N reference values, which according to the test results of the clinical samples resulted in Cq values equal to 40. Samples were determined to be negative when amplification above 40 was detected. Samples with amplification below cycle 37 indicated that the diagnosis was positive. Testing was repeated for samples with a Cq range of 37–40. A repeated Cq value less than 40 with an amplification curve that had obvious fluctuations indicated a positive result; otherwise the result was determined to be negative.

Case definition

A retested positive case was defined as one that was positive for SARS-CoV-2 in any specimen when RT-PCR tested following discharge. Recovered patients who were considered negative were those who remained negative from discharge to the end of the study period. Secondary cases were the cases resulting from a secondary transmission.

Statistical analysis

Data were double-entered using EpiData software, version 3.1 (EpiData Association, Denmark) and were analyzed using SPSS software version 11.0 (SPSS, Chicago, IL, USA). Pearson Chi-square was used to analyze and count data in different groups. The T test was used to compare the mean between groups. Tests were designed as two-tailed and an alpha value of p < 0.01 was considered to be statistically significant.

Ethical aspects

The research protocols used in this study were reviewed and approved by the Lu’an Municipal Health and Family Planning Commission and written informed consent was obtained from all research participants.

Results

Specimen nucleic acid tests

Seventeen (17/67, 25.4%) discharged COVID-19 patients retested positive for SARS-CoV-2 using RT-PCR testing. Fourteen (82.4%) were sputum positive, five (29.4%) were throat swab positive, seven (41.2%) were fecal or anal swab positive, one (5.9%) was urine sample positive, five (29.4%) were both sputum and throat swab positive, four (23.5%) were both sputum and fecal swab positive, and one (5.9%) was positive in all four specimens (Table 1).

Table 1 Results of nucleic acid tests for 17 discharged COVID-19 cases who were RT-PCR retested positive again for SARS-CoV-2.

Variables	Number ( N = 17)	Days	
Positive of specimens			
Sputum	14 (82.4%)		
Fecal or swab	7 (41.2%)		
Throat	5 (29.4%)		
Urine	1 (5.9%)		
Both sputum and throat swab	5 (29.4%)		
Both sputum and fecal	4 (23.5%)		
All four specimens	1 (5.9%)		
Frequency of positive (times)			
1	6 (35.3%)		
>1	11 (64.7%)		
Days of discharge to the last positive (days)			
Shortest		7	
Longest		48	
Median		16	
Days for collection of continuous positive (days)			
Fecal or anal swab samples		19.0 ± 10.2	
Others		14.2 ± 5.5	

Among the 17 cases, 11 (64.7%) tested positive more than once following discharge and six (35.3%) tested positive only once. The shortest amount of time between discharge and the last positive test was 7 days, the longest was 48 days, and the median interval was 16 days (Table 1).

No significant differences (p = 0.226) were observed between the average number of days required for the continued collection of positive fecal or anal swab samples (19.0 ± 10.2 days) and other specimens (14.2 ± 5.5 days) following discharge, whereas the proportion of positive fecal or anal swab samples increased from the third week (Table 1, Fig. 1).

Figure 1 Positive percentage of different types of specimens following discharge.

Serial Cq value time course for sputum samples collected from recovered positive patients on the day of illness (ORF)

Daily serial RT-PCR Cq values for the 17 recovered positive patients are shown in Fig. 2. Sputum’s median Cq value in the first week after onset was 26.7 (14.2 to 40), 37.7 (24.6 to 40) from weeks two to six, and 40 in the majority of cases after six weeks.

Figure 2 Serial Cq value time course for sputum samples collected from recovered positive patients on the day of illness (ORF).

Secondary cases

A total of 135 close contacts were tested relating to the 17 positive cases. The results of RT-PCR revealed that there were zero positive cases among those contacts.

Comparing recovered positive and unrecovered positive patient characteristics

We collected data regarding age, gender, Wuhan exposure history, underlying medical conditions, disease status, disease course, hormone use, and sputum samples’ Cq value in the early stages of the disease then performed statistical comparisons. No significant differences were noted between the groups (Table 2).

Table 2 Comparing recovered positive and unrecovered positive patient characteristics.

Variable	Recovered positive patients	Unrecovered positive patients	p value	
Age (years)	42 (3 to 66)	40.5 (1 to 78)	0.954	
Gender			0.958	
Male (n, (%))	11(25.6%)	32(74.4%)		
Female (n, (%))	6(25.0%)	18(75.0%)		
Wuhan exposure history (n, (%))	3(17.6%)	13(26.0%)	0.712	
Underlying medical conditions (n, (%))	3(17.6%)	2(4.0%)	0.188	
Disease status			0.317	
Mild and common cases (n, (%))	45(90.0%)	5(10.05)		
Severe cases (n, (%))	13(76.5%)	4(23.5%)		
Disease course (days)	22.5 ± 5.8	22.7 ± 6.8	0.901	
Hormone use [n, (%)]	3(17.6%)	14(28.0%)	0.600	
Cq value of sputum in the initial stage of the disease (ORF)	29.4(14.2 to 40)	28.8(12.7 to 40)	0.792	

Discussion

Our data demonstrate that approximately a quarter of patients with COVID-19 who recovered were found to be positive again following discharge according to the current discharge criteria in China. Most were sputum positive, throat swab positive, and fecal or anal swab positive. One patient tested positive via urine sample and only one patient was positive with all four specimens. Positive fecal or anal swab results increased from the third week onward. A recent study published in Lancet indicated that the median and longest duration of viral shedding in COVID-19 survivors was 20 and 37 days, respectively (Zhou et al., 2020a). However, we found that all of the COVID-19 cases had a positive retest less than one month from symptom onset to discharge. The discharged patients that continued to be RT-PCR positive may not have mounted an effective immune response and the infection detected by the fecal samples may require a longer time to clear (Ling et al., 2020). Our results support these findings. Therefore, it is important to assess the viral nucleic acid concentration in fecal or anal swab samples during recovery.

The seventh iteration of the COVID-19 diagnosis and treatment plan in China emphasizes the health management and isolation of patients following discharge. We identified criteria between positive patients who recovered and those who did not recover in order to aid the early screening process. However, no significant differences were noted among common epidemiological characteristics, the clinical characteristics, and the disease status, which is inconsistent with the findings of other relevant studies. Lin et al. (2021) showed that SARS-CoV-2 RNA-negative conversions are risk factors for delayed discharge from the hospital (>21 days), suggesting that there may be different discharge criteria in different regions. Our results showed no significant differences between severe, mild cases, and common cases in patients who retested positively after discharge. However, a study by Weerahandi et al. (2021) demonstrated that patients with severe cases of COVID-19 may develop health conditions several weeks after discharge, indicating that preventive measures are needed after discharge to reduce the risk of transmission in severely ill patients. Previous studies have suggested that this may be associated with host cell immunity (Ling et al., 2020). No secondary cases (cases resulting from a secondary transmission) were reported, which may be associated with the isolation measures. The Cq value of the majority of recovered patients with COVID-19 who recovered but were positive following discharge was greater than 35 within three weeks after onset, which means that infectivity may decrease significantly. The continued positive test results indicated that the possibility of viral transmission was still present and the infectious mechanism of the retested positive patients remains unknown. Viral shedding may have included the infectious virus, which should be determined in future studies. Isolation practices have been improved for COVID-19 patients in China who retested positive for viral nucleic acids following discharge (The Central People’s Government of the People’s Republic of China, 2020). However, we suggest that these discharge criteria are insufficient. SARS-COV-2 is believed to be a highly adaptable virus and South Korea reported 51 patients in April 2020 who retested positive (People’s Daily Overseas, 2020). To prevent SARS-CoV-2 transmission that may lead to new COVID-19 cases, it is essential that patients are not discharged from the hospital until they are no longer infectious. Therefore, we encourage vulnerable regions to appropriately improve their discharge criteria for COVID-19 patients. We suggest that the frequency of negative RT-PCR tests for patient samples and the time required from symptom onset or admission to discharge be increased. We also recommend expanded tests to include fecal or anal swab samples prior to discharge. As validated serology tests become available, the inclusion of tests for patient antibody levels may also be considered. If conditions permit, additional measures such as home isolation and discharge follow-up monitoring should be undertaken to ensure that no secondary cases occur. WHO’s assessment suggests that COVID-19 can be characterized as a pandemic and it is predicted that more cases will occur globally in the future (World Health Organization, WHO, 2020c, https://www.who.int/dg/speeches/detail/who-director-general-s-opening-remarks-at-the-media-briefing-on-covid-19---11-march-2020). Our hope is that we can increase the preventive measures and control the infection of this virus in China and other areas by introducing revised discharge criteria.

Our study has some deficiencies and limitations. Firstly, due to the limitations of on-site investigations, we mainly used descriptive epidemiological investigation methods and, as a result, could not explore the specific mechanism for recovering positivity in discharged patients. Secondly, due to the limited sample size, we did not find a difference in the characteristics of recovered positive and unrecovered positive patients. Therefore, a larger cohort is needed to verify our conclusions. Thirdly, virus culture tests need to be performed in laboratories that reach biosafety level 3. As we did not have access to such resources, this will be our focus in future studies.

Conclusions

We confirm that there is a high probability of testing positive after COVID-19 patients are discharged. China’s discharge criteria have remained fairly consistent so we encourage vulnerable regions to appropriately strengthen discharge criteria for COVID-19 patients, despite not being able to verify their infectivity. We hope that there will be more opportunities to conduct viral cell culture tests along with RT-PCR in biosafety level 3 laboratory conditions to better understand and verify the infectivity of COVID-19. This will help us better develop and improve the discharge criteria.

Supplemental Information

Supplemental Information 1 Raw data

The results of the analysis in the article.

Click here for additional data file.

We wish to thank the Lu’an Municipal Health and Family Planning Commission, and Lu’an Municipal People’s Hospital for their support and cooperation during our on-site investigation.

Additional Information and Declarations

Competing Interests

Author Contributions

Data Availability

The authors declare there are no competing interests.

Yong Lyu, Danni Wang and Xiude Li conceived and designed the experiments, performed the experiments, analyzed the data, authored or reviewed drafts of the paper, and approved the final draft.

Tianqi Gong, Lei Liu and Jie Sun performed the experiments, analyzed the data, authored or reviewed drafts of the paper, and approved the final draft.

Pengpeng Xu performed the experiments, authored or reviewed drafts of the paper, and approved the final draft.

The following information was supplied regarding data availability:

Raw data is available in the Supplementary File.

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
