# Peer review of "Continued nucleic acid tests for SARS-CoV-2 following discharge of patients with COVID-19 in Lu’an, China"

_PeerJ, doi:10.7717/peerj.11617_

## Round 0.1 · original submission · Major Revisions

Please take into consideration the reviewer’s comments and provide back a point-by-point rebuttal letter addressing those concerns.

Also, before submitting a revised version, please use a professional proofreading editorial service, since all reviewers and myself concur in the need of such thorough language correction.

Reviewer 1 ·

Basic reporting

1. An essential advice for authors is to consult a professional native English-speaking editing service for assistance with the manuscript, as there are several parts through it that should be carefully reviewed. Unfortunately, the manuscript is plagued with an undesirable number of spelling and grammatical errors, and poor punctuation that should be solved to improve the paper. As a consequence, some parts of the manuscript are hard to follow. Authors should understand that the reviewers´ function is to evaluate the quality and accuracy of the research presented, and provide some feedback to improve the clarity, transparency, accuracy, and utility of potential papers, not to serve as English spelling and grammar correctors.

2. Unfortunately, the study lacks a solid and clear theoretical/conceptual framework, which makes it hard to justify the study.

3. The purpose of the introduction is to orient the reader and create an interest in the study. In this manuscript, however, the introduction is too vague and makes it hard to justify it. From my perspective, there is no coherence and cohesion between the introduction and the results and conclusions to make it easier for readers to follow the authors´ arguments. Thus, authors are invited to provide a more extensive and clear introduction highlighting the importance of running a study like this. Authors should aim for a logical order that helps readers make sense of the introductory information.

Experimental design

1. The research question is ambiguous. Furthermore, the rationale for the study is not clear at all. The authors are kindly invited to provide a clear statement about the need and relevance of this study.

2. I would like to know if the research protocols used in this study were reviewed and accepted by an independent committee before initiation. The authors should mention if informed consent from research participants were given.

3. The methods need to be better described. The Methods section should give readers enough information in such a form that they can repeat the experiments. In this manuscript, several methods are unclear. Thus, the authors mention that the Sun Yat-Sen University manufactured the RT-PCR kit used, but no information about the target sequence or amplification conditions is given. In this way, interested readers may found it hard to follow the same procedure.

Validity of the findings

1. The description of the results (line 80-94) is too confusing. It is hard to follow, and data are hard to follow. The authors are invited to provide a table resuming this information.

2. From my perspective, the weakest parts of the study are the discussions and conclusions. This may be due to the lack of clarity on the results obtained.

3. The discussion is not a real discussion. The purpose of the discussion is to interpret and describe the significance of the findings in light of what has been previously published about the research problem being investigated. However, in this study, the authors cite only three studies to "discuss" their findings. Thus, the discussion lacks depth and seems biased to favor what the authors assume.

Additional comments

This could be a very interesting study if authors prepare a better manuscript. It is important to provide a scientifically supported discussion and give more clarity on methodology and results description.

I would also like the authors to look for some professional help from a native English-speaking editing service. In its actual form, several parts of the manuscript are too hard to follow.

Reviewer 2 ·

Basic reporting

Dear Authors,

The English language of your background section (lines 19 and 20) should be improved. This improvement can help an international audience clearly understand the message that you are trying to convey. For example, "four laboratory confirmed COVID-19 cases that were confirmed to be positive for SARS-CoV-2 by RT-PCR following discharges." You can improve this sentence by writing, "during their follow-up, four discharged COVID-19 individuals/patients were RT-PCR retested positive again for SARS-CoV-2." This is similar to this article: https://www.ncbi.nlm.nih.gov/pmc/articles/PMC7276353/#r2. Also, according to the PeerJ's Standard Sections (https://peerj.com/about/author-instructions/#standard-sections), subheadings should be followed by a period (e.g., Background.). Your subheadings were followed by colons (e.g., Background:, Methods:, Results:).

The English Language of your methods section (lines 22-25) should be improved as well. You can improve these sentences (lines 22-24) by writing, "We conducted RT-PCR tests on sputum, throat swab, fecal or anal swab and urine samples collected from 67 COVID-19 patients/individuals following their discharge in Lu'an Municipality, China (is this a hospital?). These samples were collected on the 7th and 14th day following their discharge." Lines 24 and 25 or the sentence, "In case the positive interval was 24 h, the samples were collected again until they were confirmed as negative for 2 times.", should be improved. If you are collecting samples on the 7th and 14th day, can you clarify the 24 hours period? Is that if a patient/individual tested positive on the 7th or 14th day, you waited 24 hours before testing them again?

The English of your results section (lines 26-32) should be improved as well.

You should have cited the following articles/reports:

https://www.ncbi.nlm.nih.gov/pmc/articles/PMC7276353/#r2

https://ophrp.org/journal/view.php?doi=10.24171/j.phrp.2020.11.3.02

https://www.cidrap.umn.edu/news-perspective/2020/05/wha-passes-pandemic-probe-resolution-korea-clarifies-reinfection-reports

https://www.frontiersin.org/articles/10.3389/fcimb.2020.00445/full

https://www.cebm.net/covid-19/infectious-positive-pcr-test-result-covid-19/

https://www.medrxiv.org/content/10.1101/2020.08.04.20167932v3 (this was published after you submitted the paper)

Experimental design

Dear Authors,

The experimental design of your manuscript should be changed. A recent systematic review by the Centre for Evidence-Based Medicine (CEBM) (https://www.cebm.net/covid-19/infectious-positive-pcr-test-result-covid-19/ ), “Are you infectious if you have a positive PCR test results for COVID-19?”, highlights the issue of using, only, PCR results to make clinical diagnosis or decisions. Jefferson et al. stated that, “PCR detection of viruses is helpful so long as its accuracy can be understood: it offers the capacity to detect RNA in minute quantities, but whether that RNA represents infectious virus may not be clear. During our Open Evidence Review of oral-fecal transmission of Covid-19, we noticed how few studies had attempted or reported culturing live SARS-CoV-2 virus from human samples.” Also, the BBC (https://www.bbc.com/news/health-54000629) wrote an article, yesterday (September 5th), about this systematic review. Furthermore, back in May of this year, “South Korean researchers revealed that discharged patients who test positive for COVID-19 again don't appear to be contagious.” This was reported by the Center for Infectious Diseases Research and Policy (CIDRAP) (https://www.cidrap.umn.edu/news-perspective/2020/05/wha-passes-pandemic-probe-resolution-korea-clarifies-reinfection-reports). The CIDRAP’s report explained that, “South Korea's investigation into recovered patients who test positive for COVID-19 again found no live virus in the patients they examined, suggesting no risk of passing the virus to another person and that the patients were likely shedding noninfectious or dead virus particles, the Korea Centers for Disease Control (KCDC) said today.” This led to a change in the KCDC’s discharge protocols (https://www.cdc.go.kr/board/board.es?mid=a30402000000&bid=0030&act=view&list_no=367267&nPage=1). According to the new KCDC’s discharge protocols, “no additional tests are required for cases that have been discharged from isolation.” Finally, their discharge protocol now states that, “the terminology for referring to such cases will be changed from “re-positive” to “PCR re-detected after discharge from isolation”.”

Your experimental design can be improved by conducting viral cell culture tests along with RT-PCR. This would be more useful to reexamine the current discharge criteria in Lu’an Municipality, china.

Validity of the findings

Dear Authors,

I think your results should have been validated using viral cell culture. Please use the paragraph bellow as a reference to improve your experimental design.

"Conclusion Prospective routine testing of reference and culture specimens are necessary for each country involved in the pandemic to establish the usefulness and reliability of PCR for Covid-19 and its relation to patients factors. Infectivity is related to the date of onset of symptoms and cycle threshold level. A binary Yes/No approach to the interpretation RT-PCR unvalidated against viral culture will result in false positives with segregation of large numbers of people who are no longer infectious and hence not a threat to public health."

https://www.medrxiv.org/content/10.1101/2020.08.04.20167932v3

Additional comments

Dear Authors,

Your manuscript is relevant to the current COVID-19 global pandemic and is in line with other publications related to COVID-19 patients testing positive after discharge. This reviewer is conscious of the tremendous amount of work and effort that went into the manuscript. Maybe you do not have access to a biosafety level 3 (BSL 3) laboratory to conduct viral cell culture experiments. However, viral cell culture experiments are important in order to accurately make clinical decisions (e.g., change discharge protocols). These decisions are crucial not only to save lives, but also to save resources.

·

Basic reporting

The article by Lyu et al., reports about the re-occurrence of positive COVID-19 tests as demonstrated by RT-PCR in 17 COVID-19 patients who had otherwise recovered and discharged from the hospital. They confirmed the same by conducting RT-PCR based testing on collect sputum, throat swab, a fecal or anal swab, and urine samples on the 7th and the 14th day following discharge. The study was conducted in a small cohort of patients n=67 in thee Lu'an Municipal Center for Disease Control and Prevention, Lu'an, China. The genes assayed for the tests were ORF and N for SARS-CoV-2.
The introduction calls for better details. Given that COVID-19 is a global pandemic, the current manuscript is very China-centric and I suggest adding a few international reports to lend more credibility to their hypothesis and conclusions. Specific details are as below
o L 43-45: Being a global pandemic the authors must include the WHO patient discharge guidelines and guidelines issued by other major nations to aid the larger global audience understand the importance of the work done.
o L 47- 51: Being a global pandemic I recommend the authors improve the information especially from outside of China at to add further rationale for their study. Since the authors explicitly mention that there have been previous reports of RT-PCR positivity in discharged patients a couple of lines on the knowledge gap being filled by the present study would enhance the read.
The language needs to be more lucid and well punctuated to avoid ambiguity. The authors may get the manuscript rewritten in proper English. There are places where the statements and observations do not match such as:
o L69-70: What do authors mean by the statement “Unrecovered positive patients were those who remained negative from discharge to the end of study observation”? I think this line should be changed to “Recovered negative patients were those who remained negative from discharge to the end of study observation.”
o L80: “A total of 17 (25.4%) out of the confirmed cases were recovered and were positive by RT-PCR” Does that mean the remaining cohort were still unrecovered and still suffering from the disease? Why were they discharged then?

Experimental design

The authors detected viral nucleic acids using RT-PCR based methodology in sputum, throat swab, fecal or anal swab, and urine samples of discharged patients.
o L58: “confirmed COVID-19 cases in Lu'an Municipality, China, who met the discharge criteria” For a wider international audience it should be properly explained what the current discharge criterion at this point rather than putting it in the discussion and the clinical, laboratory, and radiological characteristics of the patients during hospitalization who tested positive following discharge can be put in the supplement for a clear understanding of the cases.
o L61-62: Many cases of reinfection are being reported around the world a clarification on how did the authors rule out reinfection would have helped.
o L 65-66: The authors have not mentioned the validity of the assay used. Multiple assays have been designed to detect COVID-19 and most assays have a limit of detection defined. So the authors should mention the RT-PCR test kit manufacturer assay details such as the limit of detection and Ct value cut off for a positive test. Further, any RT-PCR assay has its limit of detection clearly defined, please state the same in the methods they have not stated the sensitivity of the test? Does the Ct value of 40 count as positive or simple background noise?
o A proper RT-PCR test is defined by the presence of controls. I thank you for providing the raw data, however the raw data supplied did not have any data regarding the controls and amplification curves of the tests. A true positive is always defined by a sigmoidal amplification curve.
o L75-76: the alpha value of p <0.05 was considered statistically significant, the authors should explore using a stringent statistic p <0.01 for arriving at conclusions.

Validity of the findings

o L 57-58: The cohort size of the study is very small (n=67) and only 17 were found to be having a positive RT-PCR test post-remission of all symptoms and only 1 had all samples which tested positive. For a study having implications as drastic as redesigning discharge criteria, the cohort size is too small for an effective conclusion. The authors should increase the cohort size before publishing their results.
o L110-111: Infectivity of the virus from the low viral load is yet not proved the study would have been better presented if data about the infection spreading from discharged patients would have been presented their argument and thus would have corroborated the findings and conclusions drawn. Multiple papers in the recent past have used cell line based testing to validate the presence of the infectious virus in the biological sample (e.g. Wölfel R, Corman VM, Guggemos W, et al. Virological assessment of hospitalized patients with COVID‐2019. Nature. 2020; 581:465‐469; Xiao F, Sun J, Xu Y, et al. Infectious SARS‐CoV‐2 in Feces of Patient with Severe COVID‐19. Emerg Infect Dis J. 2020; 26:1920‐1922). Wölfel et aI. 2020 did not find an infectious virus in the stool sample despite high concentrations of virus RNA, while urine samples never yielded a virus. I strongly suggest the authors conduct cell line-based studies before arriving at the conclusion they have arrived at based solely on RT-PCR testing

o L119-121: The mere possibility of viral transmission is insufficient to change discharge criterion and evidence of transmission is this regard would have greatly helped. This indicates a larger cohort would have helped in strengthening the conclusions drawn.

Additional comments

These are trying times for the scientific community in general and the biomedical scientists specifically. I want to extend my best wishes to the authors for having reported this study, however, a few points need to be kept in mind before acceptance:
1. The article in general needs to be rewritten IN PROPER LUCID ENGLISH LANGUAGE TO AVOID AMBIGUITY.
2. China-centric parameters used but a better approach would be to also USE WHO GUIDELINES while reporting patient data for wider implications and understanding of the findings.
3. For the RT-PCR, authors should mention the RT-PCR test kit assay details such as LIMIT OF DETECTION AND CT VALUE CUT OFF FOR A POSITIVE TEST.
4. The detection of viral nucleic acid does not necessarily imply that viable infectious virions are present or that the virus can spread. Also, the absence of secondary infections indicates that there was no spread. Therefore the conclusions drawn are risky, and mere extrapolation of the findings and do not truly represent the findings. CELL LINE BASED STUDY TO DETECT ACTUAL INFECTIVE VIRIONS IS A MUST BEFORE THE MANUSCRIPT IS ACCEPTED FOR PUBLICATION with the present conclusions.
5. It is difficult to work with COVID-19 samples and patients but cohort size may be expanded for better results.
Therefore I DO NOT RECOMMEND the publication IN ITS PRESENT FORM and suggest authors look into the suggestions made. The study was fairly executed but is far off when it comes to the conclusions drawn. There are some major queries as outlined above that the authors should address before the publication of this article

---

## Round 0.2 · Major Revisions

Please take into consideration the reviewer’s comments and provide back a point-by-point rebuttal letter addressing those concerns.

Please provide the certificate of a professional English language proofreading service, since the reviewer and myself still find a number of issues to be addressed.

Reviewer 1 ·

Basic reporting

Although the manuscript has substantially improved, there are still some issued that should be addressed by the authors for its publication.

1. Authors mention on their rebuttal letter that the language of their manuscript has been refined and polished by using the English Language Editing service available from PEER J Publishing. I find that difficult to believe as there are still numerous spelling and grammar mistakes. I would emphasize that the manuscript would benefit from review by a native English speaker to assist with the sentence construction and spelling. I am highlighting some suggestions on the reviewed manuscript.

Experimental design

2. In 2009, Bustin et al., proposed a number of guidelines targeting the reliability of results to help ensure the integrity of the scientific literature, promote consistency between laboratories, and increase experimental transparency. They mention that the nomenclature describing the fractional PCR cycle used for quantification is inconsistent, with threshold cycle (Ct), crossing point (Cp), and take-off point (TOP) currently used in the literature. These terms all refer to the same value from the real-time instrument and were coined by competing manufacturers of real-time instruments for reasons of product differentiation, not scientific accuracy, or clarity. Then, the authors proposed the use of quantification cycle (Cq), according to the RDML (Real-Time PCR Data Markup Language) data standard. These guidelines are highly used by groups with high quality standards on their qPCR assays. Thus, I would like to encourage the authors to adjust their manuscript to these guidelines.

Validity of the findings

3. From my perspective, it is not clear for authors what a discussion means. The purpose of the discussion is to interpret and describe the significance of the findings in light of what was already known about the research problem being investigated, and to explain any novel insights about the problem after the authors taken the findings into consideration. Thus, the discussion implies contrasting the obtained results with those reported by others. And this is the main flaw of this manuscript. The lack of a deep discussion. I would like to see the manuscript discussion go deeper by comparing their results with that of other research groups. In its current form, authors do not compare their results against other studies (for example, authors may compare their findings with those of Weerahandi et al., 2021 and Lin et al., 2021). Authors also should emphasize the relevance of their study to reinforce preventive measures to control the transmissibility of the disease.

4. Finally, and once again, the conclusions are too shallow and simplistic. The conclusion should include summarizing the paper and focus possible future studies in broader perspectives of the paper. Authors should describe how does this study has contributed to knowledge gaps (which, due to the novelty of this pandemics are abundant).

References
Bustin, S.A., Benes, V., Garson, J.A., Hellemans, J., Huggett, J., Kubista, M., Mueller, R., Nolan, T., Pfaffl, M.W., Shipley, G.L., Vandesompele, J., Wittwer, C.T. 2009. The MIQE guidelines: minimum information for publication of quantitative real-time PCR experiments. Clin Chem. 55(4):611-622.
Lin, P., Chen, W., Huang, H., Lin, Y., Cai, M., Lin, D., Cai, H., Su, Z., Zhuang, X., Yu, X. 2021. Delayed discharge is associated with higher complement C3 levels and a longer nucleic acid-negative conversion time in patients with COVID-19. Sci Rep. 11(1):1233.
Weerahandi, H., Hochman, K.A., Simon, E. et al. 2021. Post-Discharge Health Status and Symptoms in Patients with Severe COVID-19. J. Gen. Int. Med. 1-8.

Annotated reviews are not available for download in order to protect the identity of reviewers who chose to remain anonymous.

---

## Round 0.3 · accepted · Accept

Thanks for addressing all the revisions and corrections requested. Now your manuscript is accepted in PeerJ.

Reviewer 2 ·

Basic reporting

Dear Authors,

You have made significant improvements to your manuscript in terms of the English language. Also, you have added the paper (https://www.ncbi.nlm.nih.gov/pmc/articles/PMC7276353/#r2) as a reference as I previously have suggested to you. It is great to read the improved version of the manuscript.

Experimental design

Dear Authors,
I think that it improves your manuscript the fact that you stated the deficiencies and limitations of your study. For example, I have suggested that you should conduct viral cell culture experiments, but you do not have access to a biosafety level 3 laboratory. This type of information is very important for future studies or publications.

Validity of the findings

No comment

Additional comments

Dear Authors,
You have improved your manuscript and followed the suggestions and recommendations that I have previously made. This type of work is relevant to the current and ongoing COVID-19 global pandemic. This reviewer understands the hard work and effort that went into improving your manuscript. Discharge protocols for COVID-19 patients, science-based, can save lives and valuable resources (e.g., PPE, ICU beds, and oxygen) during this pandemic.